# Twisted Silica Few-Mode Hollow GeO$_2$-Doped Ring-Core Microstructured Optical Fiber

Anton V. Bourdine [1,2,3,4,*], Vladimir V. Demidov [2], Egishe V. Ter-Nersesyants [2], Grigori A. Pchelkin [2,4],
Dmitriy N. Shurupov [2,5], Alexander V. Khokhlov [2], Alexandra S. Matrosova [2], Andrey I. Kashin [2],
Sergei V. Bureev [2], Michael V. Dashkov [1], Alexander S. Evtushenko [1], Elena S. Zaitseva [1],
Azat R. Gizatulin [6], Ivan K. Meshkov [6], Amogh A. Dyavangoudar [7], Ankur Saharia [7], Manish Tiwari [7],
Alexander A. Vasilets [8], Vasiliy S. Elagin [4], Ghanshyam Singh [9] and Konstantin V. Dukelskii [2,4,10]

1  Department of Communication Lines, Povolzhskiy State University of Telecommunications and Informatics, 23, Lev Tolstoy Street, Samara 443010, Russia; mvd.srttc@gmail.com (M.V.D.); alex2194ru@yandex.com (A.S.E.); zaytzewa@inbox.ru (E.S.Z.)
2  JSC "Scientific Production Association State Optical Institute Named after Vavilov S.I.", 36/1, Babushkin Street, St. Petersburg 192171, Russia; demidov@goi.ru (V.V.D.); ter@goi.ru (E.V.T.-N.); beegrig@mail.ru (G.A.P.); shurupoff.dm@yandex.ru (D.N.S.); khokhlov@goi.ru (A.V.K.); a.pasishnik@gmail.com (A.S.M.); kashin_andrey@mail.ru (A.I.K.); bureev@goi.ru (S.V.B.); kdukel@mail.ru (K.V.D.)
3  "OptoFiber Lab" LLC, Skolkovo Innovation Center, 7, Nobel Street, Moscow 143026, Russia
4  Department of Photonics and Communication Links, Saint Petersburg State University of Telecommunications Named after M.A. Bonch-Bruevich, 22, Bolshevikov Avenue, St. Petersburg 193232, Russia; elagin.vas@gmail.com
5  Institute of Physics, Nanotechnology and Telecommunications, Peter the Great St. Petersburg Polytechnic University, BLDG. II, 29, Politekhnicheskaya Str., St. Petersburg 194064, Russia
6  Department of Telecommunication Systems, Ufa University of Science and Technology, 32, Zaki Validi Street, Ufa 450076, Russia; azat_poincare@mail.ru (A.R.G.); mik.ivan@bk.ru (I.K.M.)
7  Optoelectronics and Photonics Research Lab, Department of Electronics and Communication Engineering, Manipal University Jaipur, Jaipur 303007, India; amogh.199202014@muj.manipal.edu (A.A.D.); ankur.saharia@jaipur.manipal.edu (A.S.); manish.tiwari@jaipur.manipal.edu (M.T.)
8  Department of Physics and Mathematics Branches of Science and Information Technologies, Volga Region State University of Physical Culture, Sport and Tourism, 35, Universiade Village, Kazan 420010, Russia; a.vasilets@mail.ru
9  Department of Electronics and Communication Engineering, School of Electrical and Electronics & Communication Engineering, Malaviya National Institute of Technology, J.L.N Road, Jaipur 302017, India; gsingh.ece@mnit.ac.in
10  Faculty of Photonics and Optical Information, School of Photonics, ITMO University, BLDG. A, 49, Kronverksky Alley, St. Petersburg 197101, Russia
*  Correspondence: bourdine@yandex.ru

**Abstract:** This work presents the first instance of a silica few-mode microstructured optical fiber (MOF) being successfully fabricated with a hollow GeO$_2$-doped ring core and by strongly inducing twisting up to 790 revolutions per meter. Some technological issues that occurred during the manufacturing of the GeO$_2$-doped supporting elements for the large hollow cores are also described, which complicated the spinning of the MOFs discussed above. We also provide the results of the tests performed for the pilot samples—designed and manufactured using the untwisted and twisted MOFs described above—which were characterized by an outer diameter of 65 μm, a hollow ring core with an inner diameter of 30.5 μm, under a wall thickness of 1.7 μm, and a refractive index difference of Δn = 0.030. Moreover, their geometrical parameters, basic transmission characteristics, and the measurements of the far-field laser beam profile patterns are also provided.

**Keywords:** hollow ring-core fiber; twisted microstructured optical fiber; chirality; silica GeO$_2$-doped supporting elements; laser beam profile; laser-based few-mode optical signal transmission

## 1. Introduction

Typical hollow (or air) ring-core (HRC) optical fibers differ by their large single silica hollow cores with doped walls (or drawn from the supporting capillary, fabricated from special optical glass/material with refractive index material higher rather than pure silica) to provide a great refractive index difference between air in hollow core, core walls, and outside silica cladding, that affords unique optical waveguide properties. Nowadays, HRC microstructured optical fibers (MOFs)/photonic crystal fibers (PCFs) as well as pure HRC optical fibers are primarily targeted for the generation, maintenance, and transmission of orbital angular momentum (OAM) modes in optical telecommunication systems, based on the spatial (e.g., mode) division multiplexing (SDM/MDM) technique [1–3]. Also, this type of optical fiber has been declared to be a mode converter for differential mode delay reduction in laser-based multi-gigabit data transmission short-range optical networks with multimode optical fibers [4,5], higher-order mode dispersion compensation [5,6], and the acousto-optic control of polarization [5].

The presented work is specifically focused on the HRC optical fiber geometry described above, although there are other groups of other well-known "initial" ("basic") and "spin-off" geometries, both of which following similar fiber optic structures, containing:

-   Ring (or annular)-core optical fibers (pure silica core and cladding with the inclusion of a high refractive index material layer on the core/cladding boundary) [1–3,7–13];
-   Ring-core MOFs and PCFs (silica center, bounded by a ring from material/glass (doped silica) with a higher refractive index, and air holes in the periphery) [14,15];
-   Hollow-core MOFs and PCFs, also known as hollow-core photonic bandgap fibers (pure silica circular fiber with large air central hole (e.g., "core") and small air holes in the periphery part, placed according to desired designed geometry) [1,2,16–31];
-   Anti-resonant hollow-core fiber (AR-HCF) with complicated geometry: so-called "revolver" fibers—single/double/triple-ring AR-HCF and single/double noodles nested AR-HCF [3,29,32–34] and nodeless nested AR-HCF [3,29,32,33,35,36], "grape-fruit" [37,38], "ice-cream" [29,39].

There is a set of known published works focused on the design and simulation of HRC MOFs that guide and support the propagation of various numbers of OAM modes: 146 and 70 modes at $\lambda$ = 1100 nm and 2000 nm, respectively (HRC MOF with a total outer diameter of 116 $\mu$m and an air core of 4 $\mu$m, bounded by a phosphate optical glass ring with a wall thickness of 2 $\mu$m and the difference between the doped ring and pure silica refractive indexes of $\Delta$n = 0.11) [40], 180 OAM modes over $\lambda$ = 1500 . . . 1700 nm wavelength band (HRC MOF with a total outer diameter of 116 $\mu$m, an air core diameter of 51 $\mu$m, a ring wall thickness of 1.5 $\mu$m with $\Delta$n = 0.12 (analogue to the Schott FBG1 glass—56.7% $SiO_2$, 0.35% $Al_2O_3$, 30% PbO, 4.15% $Na_2O$, and 8.65% $K_2O$)) [41], 22 OAM modes at $\lambda$ = 1550 nm (HRC PCF with a total outer diameter of 28 $\mu$m, an air core of 11 $\mu$m, bounded by an lanthanum optical glass ring (HIKARI LaSF09) with a wall thickness of 0.1 $\mu$m and $\Delta$n = 0.37) [42], 436 at $\lambda$ = 1550 nm with 400 modes over S + C + L bands (203 $\mu$m HRC optical fiber with 100 $\mu$m hollow highly-$GeO_2$-doped-ring-core under a ring wall thickness of 1.5 $\mu$m and $\Delta$n = 0.15) [43], 874 OAM modes at $\lambda$ = 1550 nm with 514 modes over almost the entire telecommunication band (62.5 $\mu$m HRC PCF with an air core of 20 $\mu$m in diameter, bounded by the $As_2S_3$-ring with a wall thickness of 0.5 $\mu$m and an extremely high $\Delta$n = 1.00) [44], and up to 1004 OAM modes extended over the wavelength range, covering almost all ratified telecommunication bands (O, E, S, C, L), under a record-high number of 1346 OAM modes at $\lambda$ = 1550 nm (62.5 $\mu$m HRC optical fiber with 20 $\mu$m air core, bounded by $As_2S_3$-ring with a wall thickness of 0.9 $\mu$m and an extremely high $\Delta$n = 1.00) [45].

It is obvious that previously published papers, containing not only simulations, but also the main results of tests performed to successfully fabricate the designed optical fibers, are of special interest. However, in spite of the great potential for guiding and transmitting OAM modes that HRC MOFs are declared to have [1–3,40–45], there are not many reports presenting properties, the results of tests, and the measured data of manufactured HRC fibers. Finally, there is a set of works, prepared by the same group of authors, that

demonstrates the experimentally verified stable transmission of 12 OAM modes over a C-band along a designed and fabricated HRC optical fiber with an air core diameter of 6 μm, bounded by a ring with a wall thickness of 5.25 μm, and where Δn = 0.03 [3,46–49], with the following enhancing the AOM mode quantity up to 28 by enlarging the air core diameter up to 19 μm under the same ring parameters [50]. On the other hand, other "experimental" papers have presented the propagation of OAM modes over ring-core optical fibers [1–3,9,10,51–53] and hollow-core "grapefruit" MOFs [3,38,54,55]. In addition, twisted or spun MOFs and PCFs are not only declared to be a new (alternative to the primarily fiber Bragg gratings) type of fiber optic probe for sensing strain/twisting [56–58], magnetic field/electric current [59], special optical fibers for polarization maintenance, the generation of optical activity [60–65], or mode filtration [66], but they are also positioned as new optical fibers with great potentiality for guiding and transmitting OAM modes, as confirmed by not only theoretical simulations [3,67–71] but also by experimental studies [72–77]. This work presents the results of attempts to combine all the fiber optic structures described above: hollow-core MOF; ring hollow-core optical fiber; and twisted MOF. For the first time, we successfully fabricated a silica few-mode MOF with a hollow $GeO_2$-doped ring core by strongly induced twisting reaching up to 790 revolutions per meter (rpm). Some technological issues that occurred during the manufacturing $GeO_2$-doped supporting elements for large hollow cores as well as the strongly twisted MOFs described above are discussed. We provide some results of the tests, performed for pilot samples which were designed and manufactured using the untwisted and twisted MOFs described above, with an outer diameter of 67 μm; a hollow ring-core inner diameter of 25 μm, under a wall thickness of 0.85 μm, as well as specify their geometrical parameters, basic transmission characteristics, and the measurements of the far-field laser beam profile patterns.

## 2. Fiber Design

The proposed HRC MOF structure was inspired by previously designed and simulated hollow- or annular-core MOFs, destined for generating and guiding OAMs, as described in recently published papers [18,25,27,30] and especially in works [78,79], which used a similar annular-core and hexagonal geometry for the placement of the peripherical air hole. The HRC MOF pilot design presented herein is shown in Figure 1. It can be distinguished by its central hollow air core with a radius of $r_1$, bounded by a $GeO_2$-doped silica ring with an outer radius $r_2$ and a wall thickness $\Delta r = (r_2 - r_1)$. There are 108 air holes with an inner radius $r_3$ and a pith of $\Lambda$, placed over a hexagonal geometry in the periphery part of the fiber, which forms the total proposed HRC MOF structure with an outer radius of $r_4$.

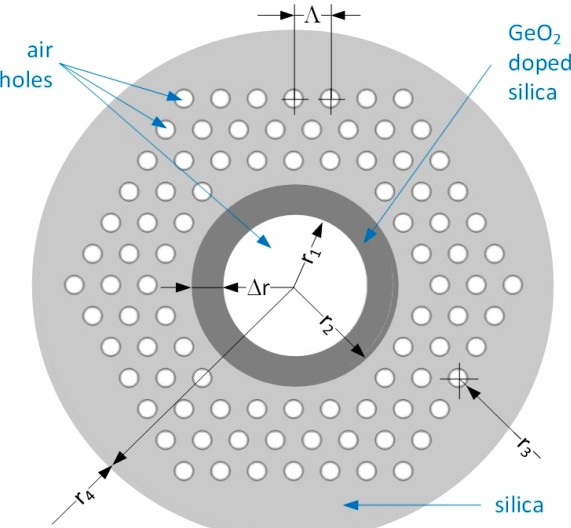

**Figure 1.** Proposed hollow $GeO_2$-doped ring-core MOF design.

The rigorous numerical full-vectorial finite-element-method by COMSOL Multiphysics® 6.1 software was utilized for the modal analysis of the designed GeO$_2$-doped HRC MOF. During the first stages, we substituted the values of radiuses $r_1 \ldots r_4$ and pitch $\Lambda$ by combining data from the specifications of the commercially available MOFs and the photonic crystal fibers and results, as presented in [18,25,27,30,78,79].

The following settings were used:

- Mesh: sequence-type physics—controlled mesh; element size—extra fine; maximum mesh element size control parameter—from study;
- Simulation: physics—electromagnetic waves; frequency domain—(ewfd); study—mode analysis;
- Study: effective mode—index; mode analysis frequency—c/$\lambda$; mode solver—ARPACK; mode search method—manual; desired number of modes—64; search for modes around—refractive index of ring core (20.5% GeO2-doped silica);
- Computation: number of degrees of freedom solved—384,935; solution time—348 s (5 min, 48 s); physical memory—4.45 Gb; virtual memory—5.02 Gb.

After several iterations of optimization, here, the following pilot parameters of the proposed HRC MOF design were finally selected: hollow-core inner radius of $r_1 = 5$ μm; GeO$_2$-doped ring wall thickness of $\Delta r = 4$ μm (HRC outer radius is $r_2 = 9$ μm); GeO$_2$-doped ring and pure silica difference of refractive indexes of $\Delta n = 0.03$; air hole radius of $r_3 = 1.7$ μm; pitch $\Lambda = 3.25$ μm. Modal analysis was performed at the wavelength $\lambda = 1550$ nm. Here, the GeO$_2$-doped ring refractive index value was estimated by the well-known Sellmeier equation [80] with the substituted coefficients, experimentally measured for GeO$_2$-SiO$_2$ glasses [81,82] under particular dopant concentrations, while the unknown value can be evaluated by a previously developed method [83].

OAM modes can be obtained by the combination of eigenmodes (*HE/EH* "even" and "odd" modes) with a $\pi/2$ phase shift, as described by the well-known summarized expressions [1–3,7–45]:

$$\begin{pmatrix} OAM_{\pm l,m}^{\pm} = HE_{l+1,m}^{even} \pm jHE_{l+1,m}^{odd} \\ OAM_{\pm l,m}^{\mp} = EH_{l-1,m}^{even} \pm jEH_{l-1,m}^{odd} \end{pmatrix}; \ (l > 1) \tag{1}$$

$$\begin{pmatrix} OAM_{\pm l,m}^{\pm} = HE_{2,m}^{even} \pm jHE_{2,m}^{odd} \\ OAM_{\pm l,m}^{\mp} = TM_{0,m} \pm jTE_{0,m} \end{pmatrix}; \ (l = 1) \tag{2}$$

It is noted that the order of the OAM mode is one order less than its respective *HE* mode, while it is reversed for *EH* modes. Therefore, two OAM modes (OAM$_{1,1}$ and OAM$_{2,1}$) are localized and supported by the proposed and simulated HRC MOF design. Their *Ez* intensity distributions, which were computed via the Poynting vector, are shown in Figure 2. Here, the OAM$_{1,1}$ mode is formed by the $HE_{2,1}$ mode, while OAM$_{2,1}$ is the result of the combination of the two vector modes $HE_{3,1}$ and $EH_{1,1}$.

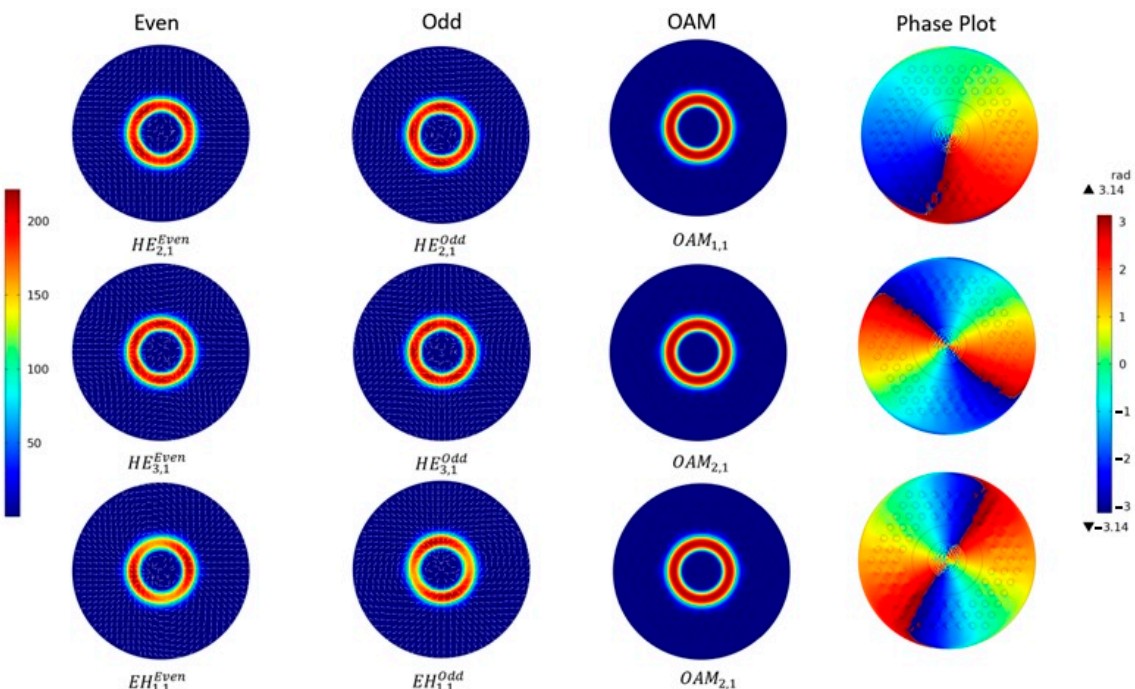

**Figure 2.** OAM mode formation by the combined *HE/EH* with the even and odd eigenmodes of the designed HRC MOF.

## 3. Fabrication of GeO₂-Doped Supporting Tubes—Preforms of Hollow Ring Core

We used the conventional modified chemical vapor deposition (MCVD) method to fabricate the silica supporting tubes with $GeO_2$-doped walls, which are preforms of the hollow ring cores. Here, the commercially available basic silica supporting tubes have the following characteristics of an outer diameter of 22 mm; a wall thickness of 2 mm; and a length of 650 mm. These were used as the basic element for the HRC preforms. These cheap supporting tubes have a high $OH^-$ group concentration (greater than 1000 ppm—which is analog to the brands Heraeus Suprasil Standard, Saint-Gobain Spectrosil A and B, Corning HPFS 7980, JGS1, Dynasil 1100/Dynasil 4100, Russian GOST 15130-86 "Optical quartz glass" KU-01, etc.), which provides the ability to perform a low-temperature MCVD-process operation that prevents tube deformation and the redundant evaporation of highly $GeO_2$-doped quartz near-surface layers.

A technological process that was developed and verified for the fabrication of $GeO_2$-doped silica tubes contains the following sequential steps:

1.  Flushing the supporting tube in distilled water with further drying under normal conditions;
2.  Installing the supporting tube in the chucks of the MCVD station.
3.  Supplying SF6 gas to the inside of the tube for the chemical etching of the distorted near-surface quartz layers;
4.  Depositing the phosphor–silicate quartz layers to prevent the diffusion of OH-groups from the supporting tube to the germane–silicate quartz layers;
5.  Depositing germane–silicate quartz layers for an improvement in the refractive index and material photosensitivity by the formation of germanium oxygen-deficient centers.

According to the aforementioned description of the technological process, we performed a series of experimentally fabricating several $GeO_2$-doped supporting tubes by further redrawing them to capillaries/HRC preforms. As a result, the optimal parameters and regimes of the technological process were successfully empirically selected:

- Delivering the rate and concentration of gas mixture/reagents (in particular, $GeCl_4$ in vapor–gas mixture to prevent bubbles, which leads to further cracks in the fabrication of the supporting element).
- Oxygen torch movement speed and its flame temperature.
- Number of torch passes.
- Dried oxygen flow rate (in $mm^3$ per minute ($mm^3$/min)), going through the bubbler systems with $GeCl_4$ and $SiCl_4$, during torch passes.
- Ratio between the numbers of phosphor–silicate quartz layers (4 . . . 9) and germane–silicate quartz layers (20 . . . 55).

During the first series of experimental fabrications, the number of torch passes that was selected was 50. However, here the supporting tube cracks, as if it was placed in the MCVD station chucks, or the increased fragility of fabricated capillaries, redrawn from the successfully deposited yet undestroyed tube, led to their collapse under further MOF drawing. After the second series of experiments, we detected that the typical dried oxygen flow rate over 400 $mm^3$/min, during its passing through the bubbler system with $GeCl_4$ and 40 . . . 50 $mm^3$/min for the $SiCl_4$ bubbler system, leads to the "boiling" of the germane–silicate quartz layers, deposited over the inner surface of the tube, or, e.g., the occurrence of bubbles in the $GeO_2$-doped layer. Therefore, during the third series of experiments, optimal regimes were empirically detected: 50 torch passes; 400 $mm^3$/min dried oxygen flow rate going through the $GeCl_4$ bubbler system; and 105 $mm^3$/min dried oxygen flow rate going through the $SiCl_4$ bubbler system. Also, we add the final deposition of five pure quartz layers to prevent the cracking of previously deposited thick germane–silicate quartz layers during the preform cooling due to the great difference between the linear expansion thermal coefficients of pure silica and those of highly doped germane–silicate quartz.

Therefore, we successfully fabricated the preforms for the $GeO_2$-doped HRCs by the proposed technique: from the supporting tube with the deposited germane–silicate quartz layers (Figure 3a,b) under the $GeO_2$-dopant concentration of 20.5 mol%, providing the desired core-cladding refractive index difference $\Delta n$ = 0.030, and further redrawing them to the capillaries.

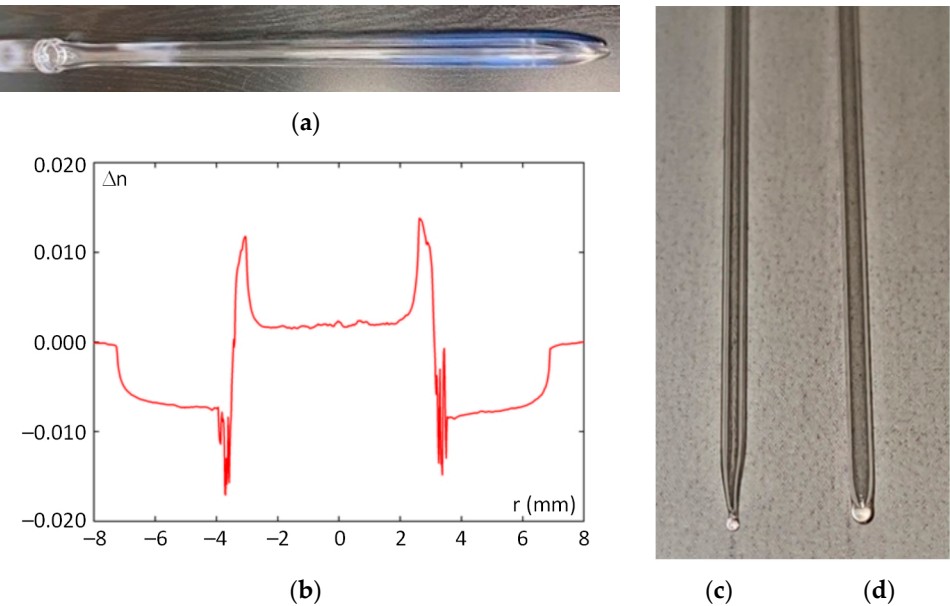

**Figure 3.** $GeO_2$-doped silica supporting tube: (**a**) tube; (**b**) supporting tube refractive index profile; (**c**,**d**) fused (closed) ends of $GeO_2$-doped capillary; (**c**) by the proposed method with splitting; and (**d**) by the conventional method.

However, during the MOF stack formation, another identical problem occurred, concerned with the cracking of the capillary, already placed into the stack. After a new series of experiments, we detected that the possible cause of capillary destruction may be explained by the fact that closed (fused) end of the capillary forms a "boule" ("ball") that extends beyond the outer diameter of the supporting element (Figure 3d). During MOF stack formation, supporting elements are densely packed in the outer main supporting tube; thus, compression between them occurs, which leads to the cracking and destruction of $GeO_2$-doped capillaries due to their increased fragility, as also explained by the aforementioned difference between the linear expansion thermal coefficients of pure silica and the highly doped germane–silicate quartz.

We developed and verified an alternative method for fusing (ending) the $GeO_2$-doped capillary, which provides a successful solution for the problem described above. Instead of the conventional positioning of the capillary end face to the side of the oxygen torch flame, we placed capillary itself with the desired length to the flame; after heating, its ends are pulled out up to splitting in the heated place, and forming tapered conical ends which are then fused (closed) (Figure 3c). As a result, the fused ends do not extend beyond the outer size of capillary, eliminating the problem of the cracking and destruction of the $GeO_2$-doped capillaries due to the compression in dense packing during MOF stack formation.

## 4. Fabrication of Silica Microstructured Optical Fibers with Improved Induced Twisting

In previously published papers [84–90], we described in detail all previous successfully performed stages of drawing the tower modifications that provide improvements to the twisting of the fabricated MOF from the initial weak 10 revolutions per meter (rpm) up to 66 rpm [84,85], and then further 100, 400, and 500 rpm [89,90]. We chose to develop the method proposed in [1]: the rotation of the optical fiber preformed during the drawing process.

During the first stage, we installed a stepper motor with a rotation speed of 200 revolutions per minute into the feed unit of the tower to add the rotation option to the drawing system, which provided a maximal induced chirality over the optical fiber of only 100 rpm under the drawing speed of 2 m per minute. We designed and fabricated a special fluoroplastic rotational adapter (Figure 4a) that directly connects an excess pressure hose to the top end of the cane and fixes it without twisting, to prevent drops in pressure between the hose and the top end of the cane due to the sealed internal space of the rotational adapter under the ability of cane rotation during the MOF drawing at the desired speed.

At the same time, we tested, verified, and determined the full-cycle technique of the twisted MOF fabrication: from the MOF stack formation, it was redrawn to the MOF cane and then followed the MOF drawing from the cane with twisting under specified excess pressure and at a specified drawing temperature [84,85]. During this stage, we successfully fabricated the following MOFs with twisting from 10 rpm (with a rotation speed of 20 revolutions per minute with an MOF drawing speed of 2 m per minute) up to 66 rpm (with a rotation speed of 200 rpm and an MOF drawing speed of 3 m per minute): hexagonal geometry with shifted core [84,85]; hexagonal geometry, that provides the quasi-ring radial mode field distribution [84,85]; equiangular spiral six-ray geometry [89,90]. Furthermore, by combining the maximal rotor rotation speed of 200 revolutions per minute and a low drawing speed of 0.9 m per minute, the typical hexagonal geometry MOF with a shifted core under the pitch of 0.65 . . . 0.700 and 3 . . . 4 spatially guided modes were manufactured with twisting of 217 rpm, which was ultimately the maximum chirality order for equipment described above that could be possibly induced.

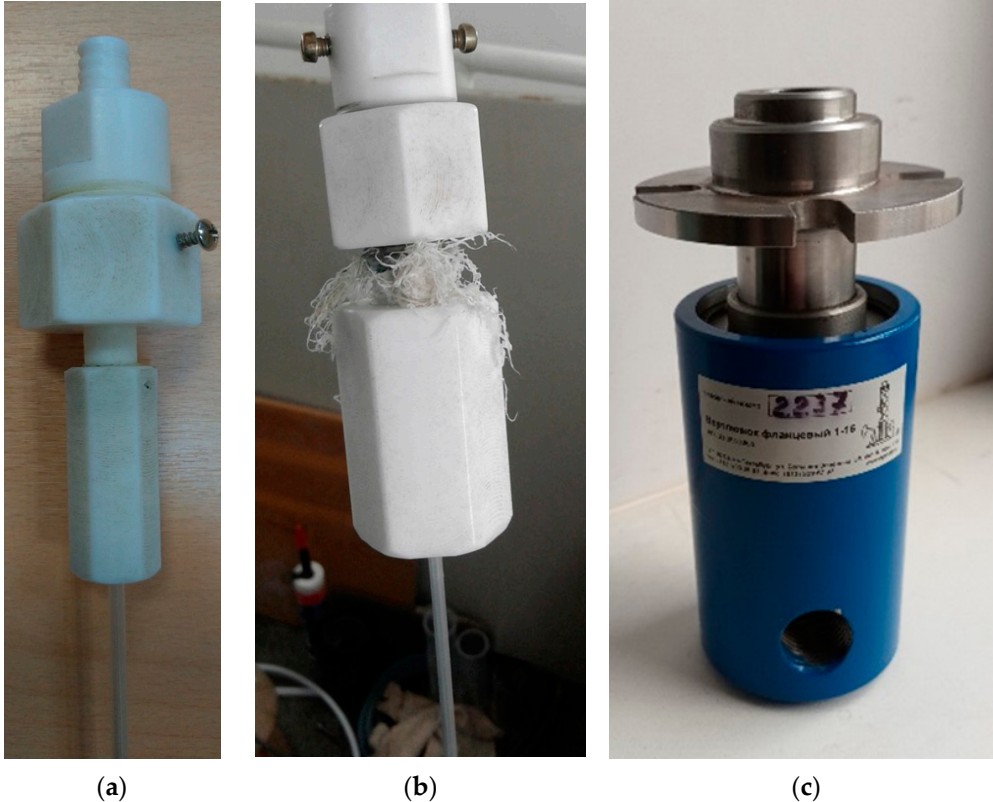

**Figure 4.** Adapters for the excess pressure feeding to the MOF cane capillaries under its rotation: (**a**) the fluoroplastic rotational adapter is completed and jointed with the installed MOF cane; (**b**) destroyed fluoroplastic adapter under an increased rotation speed of up to 600 revolutions per meter; and (**c**) commercially available flanged drilling swivel, utilized instead of the previously installed fluoroplastic adapter.

Therefore, to improve the MOF chirality, during the next stage, we replaced the stepper engine by a new commutator motor with a maximal rotation speed of 2000 revolutions per minute, and carried out a series of engineering and research and technological works on a second set of drawing tower modifications [89,90]. In addition to the design, fabrication, and installation of a new carrier for motor fixing on the feed unit chuck, a new drive belt with an improved length due to the increasing distance between the commutator motor shaft and feed unit chuck, a new special supporting pad with the fixed commutator motor, protective shroud, and driving system in the drawing tower feed unit, outside control panel, connected to power supply and driver for the commutator motor, we proposed and manufactured a special device for the additional fixing of an MOF cane to prevent its unacceptably strong vibrations in the horizontal plane with further destruction, occurring at a rotation speed of more than 300 revolutions per minute. This fixing device contains a ring stand with a clamped piece of fluoroplastic tube with a corresponding diameter, which was mounted between the tower feed unit and the tower furnace [89,90].

As a result, the performed modifications provide improvements in the preform rotation speed in the drawing tower feed unit of up to 2000 revolutions per minute, with twisting of up to 1000 rpm under a drawing speed of 2 m per minute for the silica few-mode optical fiber with a typical "telecommunication" ("coaxial") geometry (solid core, bounded by one outer solid cladding) and up to 500 rpm under the same drawing speed for MOFs.

By using the developed technique, we successfully fabricated pilot samples of silica few-mode MOFs with six $GeO_2$-doped-cores and induced twisting of 50, 100, 400, and up to 500 rpm [89,90]. However, a further increase in the rotation speed leads to the progressive destruction (abrasion) of the aforementioned fluoroplastic rotational adapter over friction surfaces (Figure 4b), which disables it and blocks the MOF cane capillaries. Therefore,

during the next stage of drawing the tower modification, we replaced the fluoroplastic rotational adapter (Figure 4a,b) by the commercially available flanged drilling swivel (Figure 4c), which provides the feeding of excess pressure to cane capillaries under a rotation speed of 500 . . . 1000 revolutions per minute. This device enables gas feeding with a pressure of up to 0.8 MPa with the desired maximal rotation speed of its spindle of 1000 revolutions per minute.

To provide the precision movement of the swivel relative to the central axis of the drawing tower furnace flame space, we designed and fabricated a special caprolon bracket, containing the swivel movement system along the *Y* axis and a device for its attachment that provides swivel movement along the *X* axis with a tight connection between the cane and swivel. Also, the special multicomponent sealed duralumin adapter tube was designed and manufactured to attach the MOF cane to the feed unit chuck. The tube was fixed by chuck jaws, which provides the rotation of the drawn MOF. As a result, the swivel with the adapter tube was mounted in the drawing tower feed unit by the caprolon bracket (Figure 5a). Here, the chuck of the feed unit is rotated by the commutator motor via belt drive, while the adapter tube is rotated via its fixation by chuck jaws (Figure 5b). The MOF cane was installed to the bottom of the adapter pipe, sub-pressed by the hold-down nut, and tightened by the silicon sealing rubber. The configuration described above enables one to feed excess pressure to the MOF cane capillaries without its uncontrolled drop over the attached unit, providing the desired chiral MOF drawing with a cane rotation speed of up to 1000 revolutions per minute (Figure 5c).

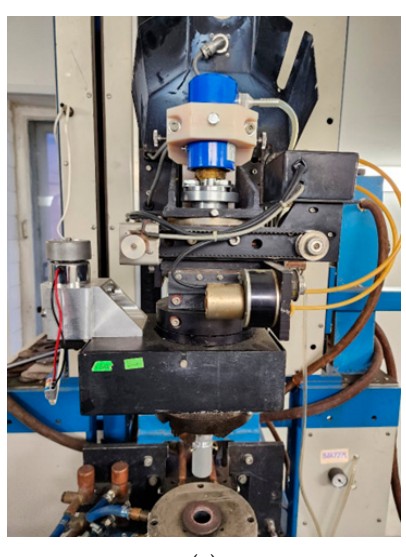 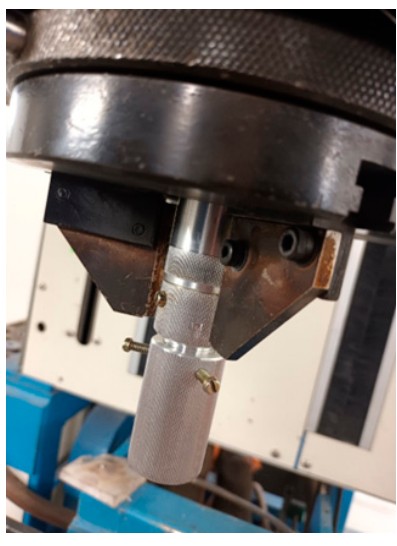 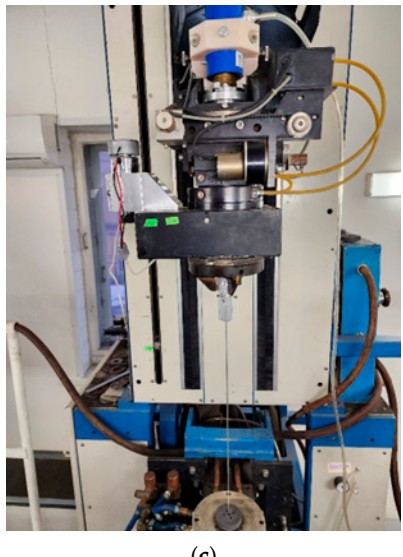

(**a**) (**b**) (**c**)

**Figure 5.** Modification of the drawing tower feed unit: (**a**) caprolon bracket with swivel and adapter tube; (**b**) adapter tube fixation by chuck jaws in the feed unit; (**c**) the whole system, prepared for the twisted MOF drawing, with an improved cane rotation speed of up to 1000 revolutions per minute.

## 5. Twisted Silica Few-Mode Hollow GeO$_2$-Doped Ring-Core Microstructure Optical Fiber: Results

We successfully fabricated two samples of the GeO$_2$-doped HRC MOF with and without inducing chirality by three typical steps of the MOF manufacturing technique [84,85,89,90]: (1) MOF stack (preform) formation; (2) cane (pre-fiber) fabrication by redrawing; and (3) the drawing of the MOF from the cane with the optionally induced twisting.

During the first step, we redrew the high-purity-fused synthetic silica supporting elements (rods and tubes with a content of hydroxyl groups (greater than 1000 ppm) to micro-rods and capillaries with an outer diameter of 1.37 mm, and cut them into segments with lengths of 30 cm. Here, we utilized cheap silica supporting tubes with a OH$^-$-group concentration (more than 1000 ppm—analog to the brands Heraeus Suprasil Standard,

Saint-Gobain Spectrosil A and B, Corning HPFS 7980, JGS1, Dynasil 1100/Dynasil 4100, Russian GOST 15130-86 "Optical quartz glass" KU-01, etc.), because the MCVD process should be performed under low temperatures to prevent tube deformation and the redundant evaporation of highly $GeO_2$-doped quartz near-surface layers, especially for a large $GeO_2$-doped HRC preparation. Therefore, after a series of operations for the treatment of both capillaries and micro-rods (end melting in the flame of an oxyhydrogen torch under blowing by dried oxygen, chemical cleaning by concentrated hydrofluoric acid, washing by distilled water, drying in a muffle furnace, etc.), we formed an MOF stack by manually placing these prepared elements inside an 18 mm inner diameter silica substrate tube (Figure 6a), according to the desired cross-section geometry/structure (Figure 1) with centrally fabricated $GeO_2$-doped HRC-preform, and melted its end by using the MCVD station.

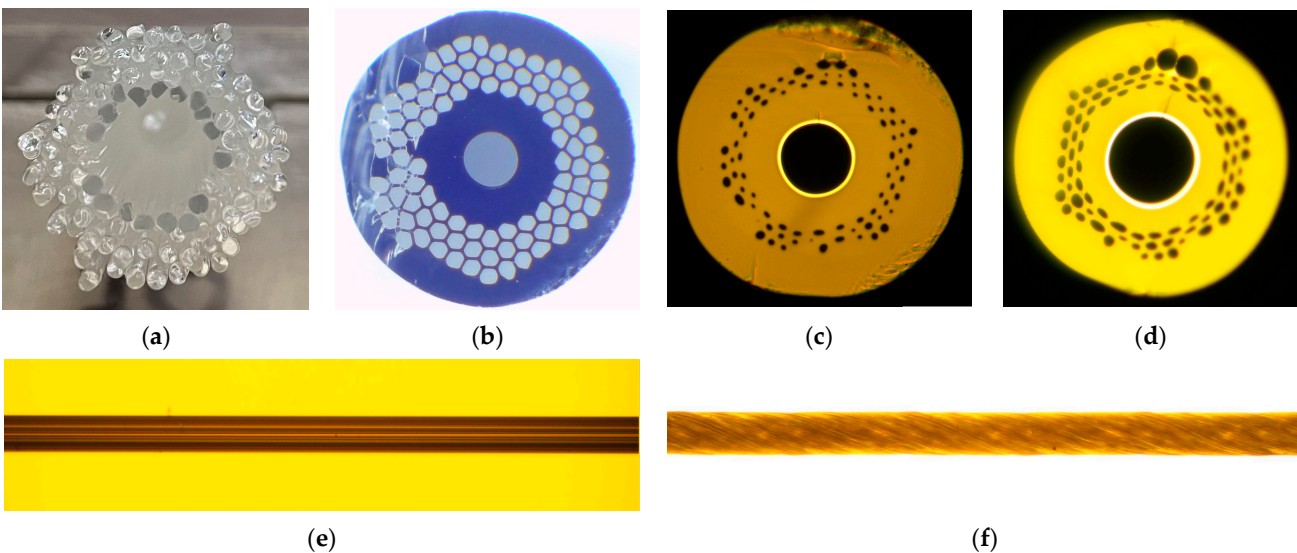

(**a**)  (**b**)  (**c**)  (**d**)

(**e**)  (**f**)

**Figure 6.** Silica $GeO_2$-doped HRC MOF: (**a**) stack; (**b**) cane; (**c**) cross-section of drawn untwisted $GeO_2$-doped HRC MOF; (**d**) cross-section of drawn $GeO_2$-doped HRC MOF with twisting 790 rpm; (**e**) longitudinal cross-section of untwisted HRC MOF; and (**f**) longitudinal cross-section of spun HRC MOF with twisting at 790 rpm.

During the second step, by using a drawing tower, we drew the MOF cane (Figure 6b) with a length of 0.5 m and an outer diameter of 3 mm from the prepared stack under vacuum at 0.5 atm, with a feeding speed of 10 mm per minute, a drawing speed of 0.5 m per minute, and at a drawing temperature of 1920 °C [82,83,87,88].

The third step consisted in the installation of the prepared MOF cane into the feed unit of the drawing tower. Then, it is driven into the high-temperature furnace and redrawn to the optical fiber under the twisting provided by the installed commutator motor with a swivel under a drawing temperature of 1900 °C, an excess pressure of 25 mbar, and an increased rotation speed of up to 1000 revolutions per minute. Therefore, we fabricated two samples of $GeO_2$-doped HRC MOF with an outer diameter of 65 μm, a hollow ring-core inner diameter of 30.5 μm, under wall thickness of 1.7 μm, and with Δn = 0.03: one untwisted (Figure 6c,e) and the other with induced twisting of 790 rpm (Figure 6d,f) with a length of about 30 m. Because the main attention was paid to HRC, most of the holes in the MOF periphery part were deformed, while the central part, containing HRC with the surrounding area, maintained its desired geometry.

The transmission spectra are presented in Figure 7. We researched the wavelength band λ = 950 ... 1700 nm by using a halogen lamp as a light source, programmable monochromator, germanium photodiode, optical amplifier, and an optical power meter. Here, the lengths of both untwisted and twisted tested HRC MOF samples was 5 m. It is noticed that, for the untwisted HRC MOF transmission wavelength range corresponding to

the short wavelengths, which is already blocked after λ = 1300 nm, the twisted HRC MOF may transfer optical emission over all the aforementioned tested wavelength band. We re-check this effect by the same tests, performed for two other MOF samples, which are cut from the opposite ends of the drawn MOFs, but these measured spectra are analogous to those in Figure 7.

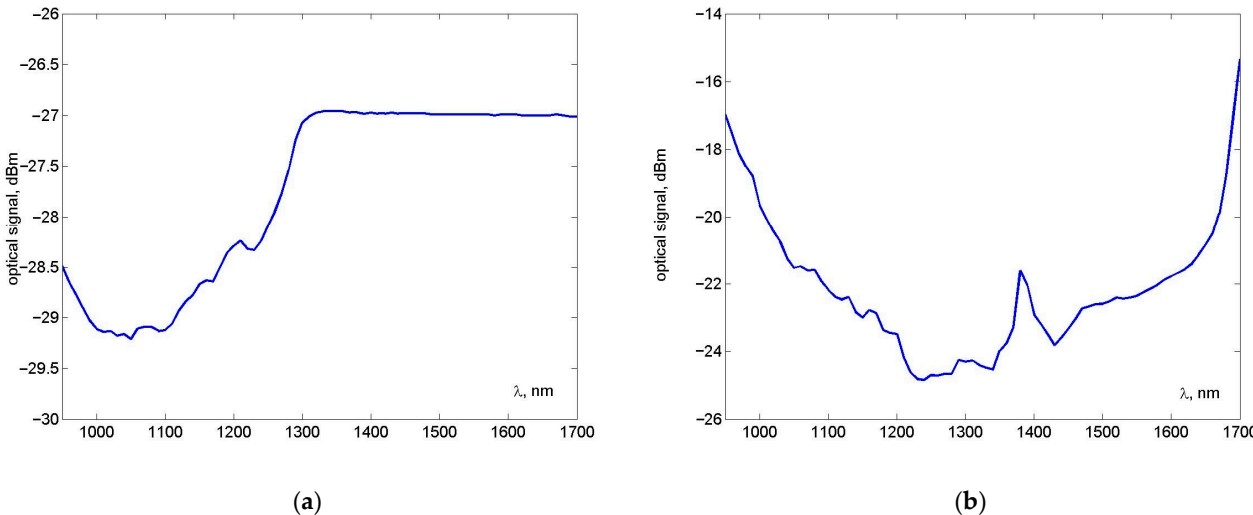

**(a)**                          **(b)**

**Figure 7.** Silica GeO$_2$-doped HRC MOF transmission spectra: (**a**) untwisted HRC MOF; and (**b**) HRC MOF with twisting 790 rpm.

The results of the first test series, concerned with the far-field laser beam profile measurements, are shown in Figure 8. Here, we utilized the "red" laser with the operation wavelength λ = 650 nm and the DFB laser with the operation wavelength λ = 1550 nm. Here, radially overfilled launching conditions via conventional multimode optical fiber (MMF) 50/125 pig-tail were provided, and it was connected to the HRC MOF via free space by the field fusion splicer precision alignment system. According to the measured images of the laser beam profile at the receiving end after its propagation over a length of 5 m of tested HRC MOF, the strongly twisted HRC MOF forms the desired "donut" structure at both λ = 650 nm and λ = 1550 nm wavelengths, while the untwisted HRC MOF provides a ring radial mode field distribution at a short wavelength of λ = 650 nm with the typical speckle pattern falling apart under the long wavelength λ = 1550 nm.

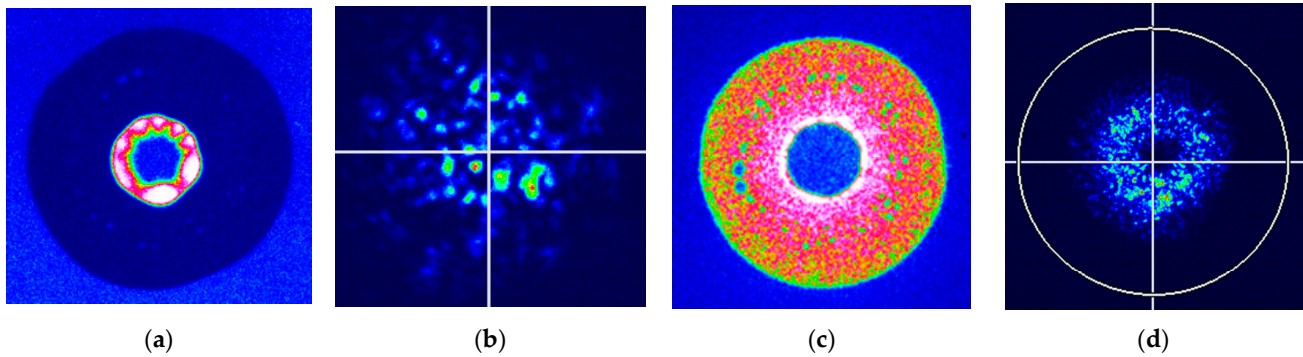

**(a)**                 **(b)**                 **(c)**                 **(d)**

**Figure 8.** Far-field laser beam profiles, measured at the receiving end after propagation over 5 m long silica GeO$_2$-doped HRC MOF under ROFL launching conditions via MMF 50/125: (**a**) untwisted HRC MOF, "red" laser λ = 650 nm; (**b**) untwisted HRC MOF, DFB laser λ = 1550 nm; (**c**) 790 rpm, "red" laser λ = 650 nm; and (**d**) 790 rpm, DFB laser λ = 1550 nm.

The next test series was also focused on the "red" laser (λ = 650 nm) beam profile measurements after propagation over the 5 m long HRC MOF. However, here we also provided centralized launching conditions via the MMF 50/125 pig-tail by fusion splicer alignment system with a varying distance between the IR-camera objective and MOF receiving end-face. Therefore, the 0 µm distance corresponds to the maximal contrast of the detected MOF end face image ("ring" due to the centralized launching conditions—Figures 9 and 10, "0 µm"), while further distances—increasing up to 100 µm—of modified laser beam pattern, and after 20 µm, saw the addition of some interferometric/optical vortex-like effects. It can be noticed that, within the mentioned distance between IR-objective and HRC MOF, the receiving end faces at least 30 µm of the untwisted HRC MOF laser beam profile consisting in a speckle pattern, while the 790 rpm twisted HRC MOF represents the stable "quasi-vortex" structure, supposedly due to the strongly improved mode coupling.

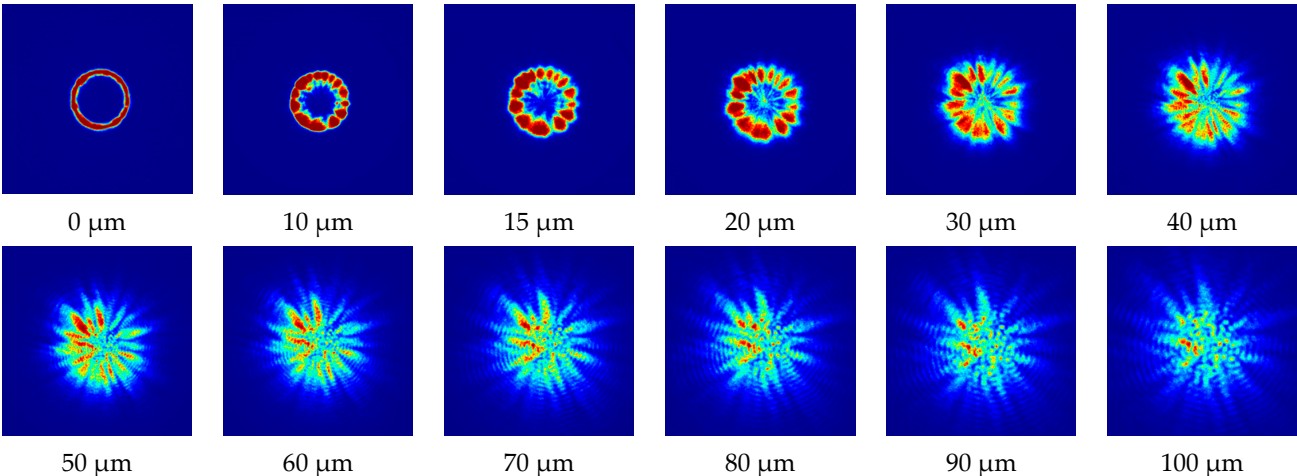

**Figure 9.** Dynamics of laser beam profile pattern, measured at the receiving end after the propagation over 5 m long untwisted HRC MOF, connected to the "red" laser (λ = 650 nm) via the MMF centralized launching conditions over the researched range of distance between the MOF receiving end and the IR-camera objective 0 . . . 100 µm.

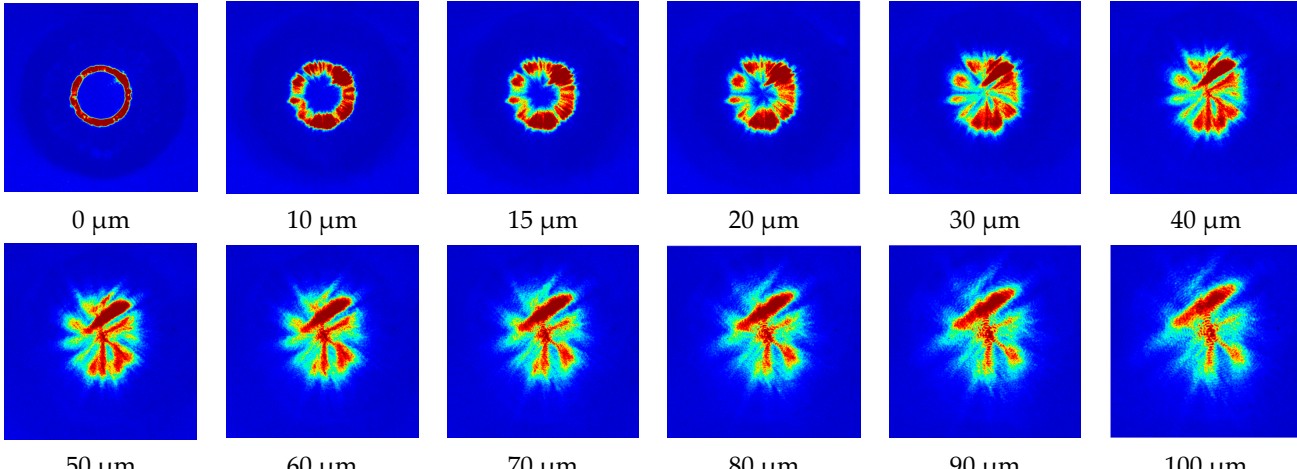

**Figure 10.** Dynamics of the laser beam profile pattern measured at the receiving end after the propagation over 5 m long 790 rpm twisted HRC MOF, connected to the "red" laser (λ = 650 nm) via the MMF centralized launching conditions over the researched range of the distance between the MOF receiving end and the IR camera objective 0 . . . 100 µm.

## 6. Conclusions

This work presents the first time that samples of silica few-mode twisted GeO$_2$-doped HRC MOF design with an outer diameter of 65 µm, an HRC inner diameter of 30.5 µm, a GeO$_2$-doped wall thickness of 1.7 µm, a refractive index difference Δn = 0.030, and a strongly induced chirality with twisting up of to 790 rpm, have been successfully fabricated. Some technological issues are discussed with regard to the fabrication of the GeO$_2$-doped supporting elements for large GeO$_2$-doped HRCs as well as of strongly twisted MOFs. We presented the results of the tests, performed with the pilot samples of the designed and fabricated— both untwisted and twisted—MOFs described above, including their geometrical parameters, basic transmission characteristics, as well as measurements of far-field laser beam profiles.

As experimentally verified herein, the strongly induced twisting of HRC MOF provides the desired "donut" structure of the radial mode field distribution superposition at both the "short" ("red" λ = 650 nm) and "long" (C-band center λ = 1550 nm) wavelengths. It can be considered a new type of fiber optic diffractive element for the mode filed management and summation/conversion. By analyzing a set of measured laser beam profiles, optimal launching conditions were detected: first of all, ROFL is required to provide the desired mode field structure ("donut" vs. "ring"). Increasing the distance between the tested HRC MOF receiving end and camera objective up to about 30 µm and more modifies and adds some interferometric effects to the beam profile pattern.

Detailed research into the presented twisted few-mode GeO$_2$-doped HRC MOF properties so that they may be utilized in measurements/sensors or/and laser systems, telecommunications, etc., requires these applications to be customized and additional series of tests and experiments in future works.

**Author Contributions:** Conceptualization, A.V.B., V.V.D., G.S., M.T., A.A.D. and A.S.; methodology, A.V.B., V.V.D., E.V.T.-N. and K.V.D.; investigation, A.V.B., V.V.D., A.V.K., A.S.M., A.I.K., G.A.P., E.V.T.-N., D.N.S., S.V.B., M.V.D., A.S.E., E.S.Z., I.K.M., A.R.G., A.A.D. and A.S.; validation, V.V.D., A.S.M., G.A.P., E.V.T.-N., D.N.S., M.V.D., E.S.Z., A.A.V., A.R.G. and I.K.M.; resources, A.V.B., K.V.D., M.V.D., A.A.V., V.S.E., M.T. and G.S.; writing—original draft, A.V.B.; writing—review and editing, A.V.B. and V.V.D.; visualization, A.V.B., A.S., A.R.G., I.K.M., G.A.P. and A.S.M.; supervision, A.V.B.; project administration, A.V.B. All authors have read and agreed to the published version of the manuscript.

**Funding:** This research received no external funding.

**Institutional Review Board Statement:** Not applicable.

**Informed Consent Statement:** Not applicable.

**Data Availability Statement:** The data presented in this study are available upon request from the corresponding author. The data are not publicly available due to the fact that they are still being collected, compared, and analyzed for the set of fabricated pilot samples of twisted MOFs with various configurations.

**Conflicts of Interest:** The authors declare no conflict of interest. The funders had no role in the design of the study; in the collection, analyses, or interpretation of the data; in the writing of the manuscript; or in the decision to publish the results.

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
