# Peer review of "Twisted Silica Few-Mode Hollow GeO2-Doped Ring-Core Microstructured Optical Fiber"

_photonics, doi:10.3390/photonics10070846_

Round 1

Reviewer 1 Report

The manuscript titled "Twisted Silica Few-Mode Hollow GeO2-Doped Ring Core Microstructured Optical Fiber" explores the intriguing findings of the successful fabrication of a silica few-mode microstructured optical fiber (MOF) with a hollow GeO2-doped ring core and strongly induced twisting up to 790 revolutions per meter. The authors discuss interesting technological issues for manufacturing GeO2-doped supporting elements. The analytical characterizations presented in this manuscript are nicely connected to each other, and the manuscript is thorough and comprehensive. Therefore, I recommend this manuscript be published in Photonics in its current form. However, I have a few specific comments that I believe would improve the manuscript.

1.      The authors should provide more details about the sequential steps involved in the fabrication of GeO2-doped silica tubes using the MCVD method.

2.      Also, they should discuss the purpose of each step, such as chemical etching, deposition of phosphor-silicate quartz layers, and deposition of germane-silicate quartz layers.

3.      The authors should explain the cause of capillary destruction during MOF stack formation in more detail. They should also explain how this problem was addressed using an alternative method for fusing the capillary ends.

4.      The authors should explain in more detail how the composition of GeO2-doped capillaries and the fusion of their ends impact their fragility and resistance to cracking during the MOF stack formation process.

5.      The authors should explain why commercially available basic silica supporting tubes with high OH-group concentration were used as the starting material for HRC performances.

6.      The authors should explain in more detail how the high OH-group concentration impacts the fabrication process.

Author Response

Reviewer 01

The manuscript titled "Twisted Silica Few-Mode Hollow GeO2-Doped Ring Core Microstructured Optical Fiber" explores the intriguing findings of the successful fabrication of a silica few-mode microstructured optical fiber (MOF) with a hollow GeO2-doped ring core and strongly induced twisting up to 790 revolutions per meter. The authors discuss interesting technological issues for manufacturing GeO2-doped supporting elements. The analytical characterizations presented in this manuscript are nicely connected to each other, and the manuscript is thorough and comprehensive. Therefore, I recommend this manuscript be published in Photonics in its current form. However, I have a few specific comments that I believe would improve the manuscript.

Many thanks for reviewing the paper and for your valuable comments. Based on your comments, the paper is revised, and the revised parts are marked by green in the revised paper.

  1. The authors should provide more details about the sequential steps involved in the fabrication of GeO2-doped silica tubes using the MCVD method.

Thank you for your comments. In Section 3 we described in details all stages of GeO2-doped silica tube (HRC preform) fabrication by MCVD method. Therefore, lines 186...197 contain general steps of fabrication, while lines 198…210 name parameters and regimes of technological process, which was empirically determined after a series of preliminary attempts. Lines 211…225 describe details of the process (including number of torch passes, flow rate etc.).

  1. Also, they should discuss the purpose of each step, such as chemical etching, deposition of phosphor-silicate quartz layers, and deposition of germane-silicate quartz layers.

Thank you for your comment. Please find purposes of each step in lines 186…197. Please note, that the most of them are typical for well-known MCVD-process (like flushing of tubes, chemical etching by SF6 gas or deposition of phosphor-silicate quartz layers). Therefore, we did not overload the paper by description of well-known technical operations. However, in lines 186…197 we named them and noted their purposes.

  1. The authors should explain the cause of capillary destruction during MOF stack formation in more detail. They should also explain how this problem was addressed using an alternative method for fusing the capillary ends.

Thank you for your comment. Please find explanation in lines 234…242. So, typical technique for fusing (e.g. closing) of capillary end produces “boule” (“ball”) with size larger, than outer diameter of capillary. This leads to unwanted pressure / stress for GeO2-doped MOF supporting elements and to cracking due to non-equal linear expansion thermal coefficients in comparison with pure silica. Therefore, we proposed to close (to fuse) capillary by alternative method, that provides tapering of its end, and solve the problem of unwanted large (greater, than outer diameter of capillary) size “boule” formation.

  1. The authors should explain in more detail how the composition of GeO2-doped capillaries and the fusion of their ends impact their fragility and resistance to cracking during the MOF stack formation process.

Thank you for your comment. Please see the answer on previous comment / lines 234…242 (explanation of the problem) and 243…250 (description of proposed solution). Please note, that this problem occurred only for MOF stacks, containing GeO2-doped supporting elements, while for pure silica MOF stacks (silica MOFs) it is not present (we associate it with difference between linear expansion thermal coefficients). Therefore, this “cracking” problem was successfully solved by “tapering” ends before closing – so, we utilized this procedure for all “annular” supporting elements – both pure silica and GeO2-doped micro-tubes.

  1. The authors should explain why commercially available basic silica supporting tubes with high OH-group concentration were used as the starting material for HRC performances.

Thank you for your comment. Please find the answer in lines 176…185. So, the main reasons to utilize supporting tubes with OH-group concentration are just low price and their market availability in Russia, while high quality “dried” quartz tubes for “all-wave” optical fibers with extremely low OH-concentration are much more expensive under the problems with supplement due to sanctions. Because we planned (and performed) a set of series for pilot experimental fabrications from GeO2-doped HRC preformed up to final twisted GeO2-doped HRC MOF, we decided to utilize cheap and available supporting tubes. Also, OH-group concentration provides ability for low temperature MCVD-process operation to avoid tube deformation and redundant evaporation of highly GeO2-doped quartz near-surface layers.

  1. The authors should explain in more detail how the high OH-group concentration impacts the fabrication process.

Thank you for your comment. Please find the answer in lines 183…185: OH-group concentration provides ability for low temperature MCVD-process operation to avoid tube deformation and redundant evaporation of highly GeO2-doped quartz near-surface layers.

Reviewer 2 Report

The authors designed and successfully fabricated a few-mode microstructured optical fiber with hollow GeO2-doped ring core and strongly induced twisting. Some technological issues were discussed. The fiber samples were characterized. The results were supported by the finite-element simulation. The developed fiber construction is sufficiently novel and interesting for different tasks. I have only a few comments for the authors.

1. Please comment on how the geometric parameters of the fibers were chosen (Fig. 1)? There are a lot of parameters and it is not clear if there are preferences from a technological or physical point of view.

2. It would be useful to add a colorbar for phases in Fig. 2. It should be also mentioned what is shown in Fig. 2 as modes (the Poynting vector or |Ex|^2+|Ey|^2, or something else).

3. Cross-section of the twisted fiber in Fig. 6d seems to be more regular than the cross-section of the untwisted fiber in Fig. 6c. Please comment this fact.

Author Response

Reviewer 02

The authors designed and successfully fabricated a few-mode microstructured optical fiber with hollow GeO2-doped ring core and strongly induced twisting. Some technological issues were discussed. The fiber samples were characterized. The results were supported by the finite-element simulation. The developed fiber construction is sufficiently novel and interesting for different tasks. I have only a few comments for the authors.

Many thanks for reviewing the paper and for your valuable comments. Based on your comments, the paper is revised, and the revised parts are marked by blue in the revised paper.

  1. Please comment on how the geometric parameters of the fibers were chosen (Fig. 1)? There are a lot of parameters and it is not clear if there are preferences from a technological or physical point of view.

Thank you for your question. We have added to Section 2 brief description and comments:

Proposed HRC MOF structure was inspired by earlier on designed and simulated hollow or annular core MOFs, targeted for OAM generation and guiding, described in recently published papers [18, 25, 27, 30] and especially by works [78, 79] with similar annular core and hexagonal geometry of periphery air hole placement. Presented HRC MOF pilot design is shown on Figure. 1. It differs by central hollow air core with radius r1, bounded by GeO2-doped silica ring with outer radius r2 and wall thickness Dr = (r2 r1). There are 108 air holes with inner radius r3 and pith L, placed over hexagonal geometry in the periphery part of fiber, which forms total proposed HRC MOF structure with outer radius r4.

Rigorous numerical full-vectorial Finite-Element-Method by COMSOL Multiphysics® software for modal analysis of designed GeO2-doped HRC MOF was utilized. During the first stages, we substituted values of radiuses r1…r4 and pitch L by combining data from specifications of commercially available MOFs and photonic crystal fibers and results, presented in [18, 25, 27, 30, 78, 79].

Rigorous numerical full-vectorial Finite-Element-Method by COMSOL Multiphysics® software for modal analysis of designed GeO2-doped HRC MOF was utilized. During the first stages, we substituted values of radiuses r1…r4 and pitch L by combining data from specifications of commercially available MOFs and photonic crystal fibers and results, presented in [18, 25, 27, 30, 78, 79].

Following settings were used:

  • mesh: sequence type physics-controlled mesh; element size – extra fine; maximum mesh element size control parameter – from study;
  • simulation: physics – electromagnetic waves, frequency domain (ewfd); study – mode analysis;
  • study: effective mode index; mode analysis frequency – c/λ; mode solver – ARPACK; mode search method – manual; desired number of modes – 64; search for modes around – refractive index of ring core (20.5% GeO2 doped silica);
  • computation: number of degrees of freedom solved for: 384935; solution time: 348 s. (5 minutes, 48 seconds); physical memory: 4.45 Gb; virtual memory: 5.02 Gb.

 After several iterations of optimization, here the following pilot parameters of proposed designed HRC MOF were finally selected:  hollow core inner radius r1=5 µm; GeO2-doped ring wall thickness Dr=4 µm (HRC outer radius is r2=9 µm); GeO2-doped ring and pure silica difference of refractive indexes Dn=0.03; air hole radius r3=1.7 µm; pitch L=3.25 µm. Modal analysis was performed at the wavelength l=1550 nm. Here GeO2-doped ring refractive index value was estimated by well-known Sellmeier equation [80 et al.] with substituted coefficients, experimentally measured for GeO2-SiO2 glasses [81, 82] under particular dopant concentrations, while unknown one can be evaluated by earlier on developed method [83].

  1. It would be useful to add a colorbar for phases in Fig. 2. It should be also mentioned what is shown in Fig. 2 as modes (the Poynting vector or |Ex|^2+|Ey|^2, or something else).

Thank you for your comment and question. We have added to the Figure 2 colorbars and corresponding comment to Section 2.

It is noted, that the order of OAM mode is one order less, than its respective HE mode, while it is reversed for EH modes. Therefore, two OAM modes (OAM­1,1 and OAM2,1) are localized and supported by proposed designed and simulated HRC MOF. Their Ez intensity distributions, been computed via Poynting vector, are shown on Figure 2. Here, OAM1,1 mode is formed by HE2,1 mode, while OAM2,1 is the result of combination of two vector modes HE3,1 and EH1,1.

  1. Cross-section of the twisted fiber in Fig. 6d seems to be more regular than the cross-section of the untwisted fiber in Fig. 6c. Please comment this fact.

Many thanks for your remark. First of all, please note, that in the first version of paper we have placed images of two MOFs, drawn from two various preforms under non-identical parameters of drawing process (drawing speed, temperature, excessive pressure etc.). To correct the described mistake, we replaced Figures 6 (c) and (d) to photos with MOF end face images, been drawn from the same MOF preform under the same parameters of drawing process. Here only excessive pressure should be improved, during the drawing of strongly twisted MOF in comparison with untwisted MOF. It may be considered as the main reason, that air holes of twisted MOF are some larger and deformed in comparison with untwisted MOF, where some of holes, on the contrary, were collapsed. This fact may be explained by lower excessive pressure, which was used during the drawing of untwisted HRC MOF to keep the HRC circularity, while technological process of drawing HRC MOF with induced strong twisting initially requires improved excessive pressure.

Reviewer 3 Report

This work fabricated silica few-mode microstructured optical fiber (MOF) with hollow GeO2-doped ring core. Some technological issues for manufacturing of GeO2-doped supporting elements for large hollow cores as well as described above complicated spun MOFs are discussed. The theoretical analysis and experimental results are complete. Overall, I recommend it for publication, provided the following concerns are addressed. 

1、   In the introduction, it seems that the research motivation of this paper is not clearly described. It was mentioned that few researcher was doing it, but it was not clear what specific problems it would solve or improve (such as performance metrics).

2、   The concept of few mode appears very suddenly, but it appears in the title and key words. I think it is necessary for the authors to add some information about the few mode in the introduction.

3、   In “2. Fiber Design”, why propose a structure with such parameters? Was it derived from theoretical derivation or from multiple simulations?

4、   It needs to specify the mesh size and the time step for the numerical simulation.

5、   A comparative and conclusive description of Figure 2 is required.

6、   In “3. Fabrication of XXX”, what are the advantages of the modified chemical vapor deposition method compared to other methods? Does the size and dimensions of the silica supporting tubes affect the experiment?

7、   The authors need to compare the proposed silica few-mode microstructured optical fiber with other similarly structured fibers in order to highlight its advantages and disadvantages.

Author Response

Reviewer 03

This work fabricated silica few-mode microstructured optical fiber (MOF) with hollow GeO2-doped ring core. Some technological issues for manufacturing of GeO2-doped supporting elements for large hollow cores as well as described above complicated spun MOFs are discussed. The theoretical analysis and experimental results are complete. Overall, I recommend it for publication, provided the following concerns are addressed. 

Many thanks for reviewing the paper and for your valuable comments. Based on your comments, the paper is revised, and the revised parts are marked by yellow in the revised paper.

  1. In the introduction, it seems that the research motivation of this paper is not clearly described. It was mentioned that few researchers were doing it, but it was not clear what specific problems it would solve or improve (such as performance metrics).

Thank you for your comment. Please find the response in lines 115…118. We proposed to combine two basic approaches – HRC MOF and strongly twisted MOF. Both fiber optic structures are declared as OAM guiding systems. So, it was supposed, that combination of these two structures may improve effects of OAM management. First time, we successfully fabricated silica few-mode MOF with hollow GeO2-doped ring core and strongly induced twisting up to 790 revolutions per meter and presented some results of experimental researches / measurements of its basic parameters.

  1. 2.The concept of few mode appears very suddenly, but it appears in the title and key words. I think it is necessary for the authors to add some information about the few mode in the introduction.

Thank you for your comment. Concerning with few-mode regime, the most part of introduction is focused on known solutions of MOFs for OAM mode generation and guiding.  It is well known, that OAM modes are obtained by combination of eigenmodes (HE / EH “even” and “odd” modes) with a pi/2 phase shift. Therefore, OAM mode is initially product of two eigen mode summation, it is directly few-mode effect, and transmission of optical signal by OAM modes requires few-mode operation of optical fiber.

  1. 3.In “2. Fiber Design”, why propose a structure with such parameters? Was it derived from theoretical derivation or from multiple simulations?

Thank you for your question. We have added to Section 2 brief description and comments:

Proposed HRC MOF structure was inspired by earlier on designed and simulated hollow or annular core MOFs, targeted for OAM generation and guiding, described in recently published papers [18, 25, 27, 30] and especially by works [78, 79] with similar annular core and hexagonal geometry of periphery air hole placement. Presented HRC MOF pilot design is shown on Figure. 1. It differs by central hollow air core with radius r1, bounded by GeO2-doped silica ring with outer radius r2 and wall thickness Dr = (r2 r1). There are 108 air holes with inner radius r3 and pith L, placed over hexagonal geometry in the periphery part of fiber, which forms total proposed HRC MOF structure with outer radius r4.

Rigorous numerical full-vectorial Finite-Element-Method by COMSOL Multiphysics® software for modal analysis of designed GeO2-doped HRC MOF was utilized. During the first stages, we substituted values of radiuses r1…r4 and pitch L by combining data from specifications of commercially available MOFs and photonic crystal fibers and results, presented in [18, 25, 27, 30, 78, 79]. After several iterations of optimization, here the following pilot parameters of proposed designed HRC MOF were finally selected:  hollow core inner radius r1=5 µm; GeO2-doped ring wall thickness Dr=4 µm (HRC outer radius is r2=9 µm); GeO2-doped ring and pure silica difference of refractive indexes Dn=0.03; air hole radius r3=1.7 µm; pitch L=3.25 µm. Modal analysis was performed at the wavelength l=1550 nm. Here GeO2-doped ring refractive index value was estimated by well-known Sellmeier equation [80 et al.] with substituted coefficients, experimentally measured for GeO2-SiO2 glasses [81, 82] under particular dopant concentrations, while unknown one can be evaluated by earlier on developed method [83].

  1. 4.It needs to specify the mesh size and the time step for the numerical simulation.

Thank you for your valuable remark. We have added description in Section 2.

Following settings were used:

  • mesh: sequence type physics-controlled mesh; element size – extra fine; maximum mesh element size control parameter – from study;
  • simulation: physics – electromagnetic waves, frequency domain (ewfd); study – mode analysis;
  • study: effective mode index; mode analysis frequency – c/λ; mode solver – ARPACK; mode search method – manual; desired number of modes – 64; search for modes around – refractive index of ring core (20.5% GeO2 doped silica);
  • computation: number of degrees of freedom solved for: 384935; solution time: 348 s. (5 minutes, 48 seconds); physical memory: 4.45 GB; virtual memory: 5.02 Gb.

  1. A comparative and conclusive description of Figure 2 is required.

Thank you for your valuable remark. We have added comments to Figure 2.

Therefore, two OAM modes are localized and supported by proposed designed and simulated HRC MOF. It is noted, that the order of OAM mode is one order less, than its respective HE mode, while it is reversed for EH modes. Therefore, two OAM modes (OAM­1,1 and OAM2,1) are localized and supported by proposed designed and simulated HRC MOF. Their Ez intensity distributions, been computed via Poynting vector, are shown on Figure 2. Here, OAM1,1 mode is formed by HE2,1 mode, while OAM2,1 is the result of combination of two vector modes HE3,1 and EH1,1.

  1. 6. In “3. Fabrication of XXX”, what are the advantages of the modified chemical vapor deposition method compared to other methods? Does the size and dimensions of the silica supporting tubes affect the experiment?

Thank you for your question. The main reason, that we utilized MCVD-method is just, that R&D lab of JSC “SPA SOI” is equipped by only MCVD-stations. We do not have PCVD-station, that provides the most preferable technique for high-precision deposition of dopant layers. Concerning the second question about size and dimensions of supporting tubes affecting on the experiment, we utilized low price commercially available silica supporting tubes with outer diameter 22 mm, wall thickness 2 mm and length 650 mm (please, see lines 176…185). Then we redraw them to capillaries with outer diameter 1.37 mm. The same matter was for silica rods. About the HRC – here, because we have results of pilot HRC MOF simulation, we tried to prepare MOF stack by taking into account desired both HRC and total MOF dimensions. Therefore, finally, we fabricated and researched only two same fiber optic structures (HRC MOFs with the same geometry / dimensions), differing only by induced twisting.

  1. The authors need to compare the proposed silica few-mode microstructured optical fiber with other similarly structured fibers in order to highlight its advantages and disadvantages.

Thank you for your comment. Please note, that this work presents new, first time successfully fabricated complicated optical waveguide structure – twisted GeO2-doped HRC MOF. It differs by strong twisting (up to 790 revolutions per meter) under inclusion to structure large hollow GeO2-doped ring core. That is why we paid attention to the following main topics, containing the scope of paper:

  1. Detailed description of developed technique for the first time in Russia fabrication of strongly twisted MOFs – from performed modifications of drawing tower to selected optimal manufacturing (technological) process regime (temperature, excessive pressure, speed of drawing, speed of preform rotation etc.).
  2. Detailed description of developed technique for fabrication of silica GeO2-doped MOF supporting elements – micro-tubes / capillaries (e.g. HRC preform).
  3. Engineering analysis of designed GeO2-doped HRC MOF, performed by COMSOL, to detect GeO2-dopant concentration in HRC and all MOF configuration, providing 2 OAM modes.
  4. Presentation of new (fabricated first time) twisted GeO2-doped HRC MOF.
  5. Pilot tests of fabricated lengths of described untwisted and twisted GeO2-doped HRC MOFs – here the main aim was just to confirm its operation / ability to transmit light / optical emission – that is why we researched only spectral responses and laser beam profiles.

Of course, following series of both simulation and experimental researches should be performed to produce adequate comparison with other known solutions. In this paper we confirmed ability of fabrication of complicated optical fiber, that combines two basic configuration – HRC MOF and twisted MOF.

Reviewer 4 Report

The authors demonstrate twisted optical silica fiber with small propagating modes. This work has some interesting in the field of microstructure optical fiber. Some comments should be revised before considering the acceptance of this manuscript.

1)     The detailed parameters of device simulation based on COMSOL could be added the manuscript.

2)     How about the repeatability of this twisted fiber device?

3)     What about the advantage of this twisted fiber relative to those existed references. Please give a comparison.

4)     The related important reference (Optics Letters, 2020, 45(14), 3889-3892) could be added in the updated manuscript.

5)     In Fig. 6, one can observe that the transverse interface is not uniform. Whether the optical spectrum could be influenced by these defects.

Author Response

Reviewer 04

The authors demonstrate twisted optical silica fiber with small propagating modes. This work has some interesting in the field of microstructure optical fiber. Some comments should be revised before considering the acceptance of this manuscript.

Many thanks for reviewing the paper and for your valuable comments and questions. Based on your comments, the paper is revised, and the revised parts are marked by red in the revised paper.

1)     The detailed parameters of device simulation based on COMSOL could be added the manuscript.

Thank you for your valuable comment. We have added description in Section 2.

Following settings were used:

  • mesh: sequence type physics-controlled mesh; element size – extra fine; maximum mesh element size control parameter – from study;
  • simulation: physics – electromagnetic waves, frequency domain (ewfd); study – mode analysis;
  • study: effective mode index; mode analysis frequency – c/λ; mode solver – ARPACK; mode search method – manual; desired number of modes – 64; search for modes around – refractive index of ring core (20.5% GeO2 doped silica);
  • computation: number of degrees of freedom solved for: 384935; solution time: 348 s. (5 minutes, 48 seconds); physical memory: 4.45 GB; virtual memory: 5.02 Gb.

2)     How about the repeatability of this twisted fiber device?

Thank you for your question. After series of drawing tower modifications (please, see the Section 4), we may draw repeatable twisted MOFs with length of up to 50 m with induced twisting 790 rpm (revolutions per meter) under stable geometry. Reducing of twisting down to 500 rpm removes restrictions on MOF length drawing – finally it was limited only by length of stack / MOF preform. We successfully fabricated long weakly and mid twisted MOFs (from 100 rpm up to 500 rpm) with stable geometry with lengths more 100 m.

3)     What about the advantage of this twisted fiber relative to those existed references. Please give a comparison.

Thank you for your comment. Please note, that this work presents new, first time successfully fabricated complicated optical waveguide structure – twisted GeO2-doped HRC MOF. It differs by strong twisting (up to 790 revolutions per meter) under inclusion to structure large hollow GeO2-doped ring core. That is why we paid attention to the following main topics, containing the scope of paper:

  1. Detailed description of developed technique for the first time in Russia fabrication of strongly twisted MOFs – from performed modifications of drawing tower to selected optimal manufacturing (technological) process regime (temperature, excessive pressure, speed of drawing, speed of preform rotation etc.).
  2. Detailed description of developed technique for fabrication of silica GeO2-doped MOF supporting elements – micro-tubes / capillaries (e.g. HRC preform).
  3. Engineering analysis of designed GeO2-doped HRC MOF, performed by COMSOL, to detect GeO2-dopant concentration in HRC and all MOF configuration, providing 2 OAM modes.
  4. Presentation of new (fabricated first time) twisted GeO2-doped HRC MOF.
  5. Pilot tests of fabricated lengths of described untwisted and twisted GeO2-doped HRC MOFs – here the main aim was just to confirm its operation / ability to transmit light / optical emission – that is why we researched only spectral responses and laser beam profiles.

Of course, following series of both simulation and experimental researches should be performed to produce adequate comparison with other known solutions. In this paper we confirmed ability of fabrication of complicated optical fiber, that combines two basic configuration – HRC MOF and twisted MOF.

4)     The related important reference (Optics Letters, 2020, 45(14), 3889-3892) could be added in the updated manuscript.

Thank you for your comment. Presented paper introduction is focused on optical fiber solutions (especially, various configuration of MOFs) for OAM generation and guiding. The reference Huibo Fan, Wenwen Ma, Liang Chen, and Xiaoyi Bao “Ultracompact twisted silica taper for 20 kHz to 94 MHz ultrasound sensing” (Optics Letters, 2020, 45(14), 3889-3892) is out of paper topic.

5)     In Fig. 6, one can observe that the transverse interface is not uniform. Whether the optical spectrum could be influenced by these defects.

Thank you for your question. First of all, please note, that in the first version of paper we have placed images of two MOFs, drawn from two various preforms under non-identical parameters of drawing process (drawing speed, temperature, excessive pressure etc.). To correct the described mistake, we replaced Figures 6 (c) and (d) to photos with MOF end face images, been drawn from the same MOF preform under the same parameters of drawing process. Here only excessive pressure should be improved, during the drawing of strongly twisted MOF in comparison with untwisted MOF. It may be considered as the main reason, that air holes of twisted MOF are some larger and deformed in comparison with untwisted MOF, where some of holes, on the contrary, were collapsed. This fact may be explained by lower excessive pressure, which was used during the drawing of untwisted HRC MOF to keep the HRC circularity, while technological process of drawing HRC MOF with induced strong twisting initially requires improved excessive pressure.

Concerning to transmission spectra, we re-check and experimentally confirmed effect of blocking the optical emission on long wavelengths in untwisted HRC MOF for various samples of fibers (drawn from various preforms) by two independent research teams (JSC “SPA SOI” and PSUTI). We suppose, this effect occurs due to spectral response was measured via single mode optical fiber, connected to tested MOF via free space with air gap to provide under overfilled launching conditions. After 1300 nm launched emission to untwisted HRC MOF starts to un-satisfy cut-off / emission guiding conditions. While for strongly twisted HRC MOF improved mode coupling should be noticed. Also, earlier on we experimentally verified, that inclusion to MOF structure silica GeO2-doped supporting elements under MOF twisting order improvement will provide contraction (e.g., fo-cusing) of radial mode field distribution to core centers (Bourdine A.V. et al. Fibers. – 2023. – vol. 11(3). – P. 28-1 – 28-18; https://doi.org/10.3390/fib11030028 https://www.mdpi.com/2079-6439/11/3/28). It may be considered as the reason and experimental confirmation, that induced strong twisting affects on both transmission spectra and laser beam profile pattern.

Round 2

Reviewer 4 Report

I have no comments, and suggest the manuscript to be published in Photonics.